# Seeing the Image: Prioritizing Visual Correlation by Contrastive Alignment

**Xin Xiao**[1,2*†]**, Bohong Wu**[2*]**, Jiacong Wang**[2,3†]**, Chunyuan Li**[2]**, Xun Zhou**[2]**, Haoyuan Guo**[2]

[1]School of Computer Science, Wuhan University [2]ByteDance Inc.
[3]School of Artificial Intelligence, University of Chinese Academy of Sciences
https://github.com/foundation-multimodal-models/CAL

## Abstract

Existing image-text modality alignment in Vision Language Models (VLMs) treats each text token equally in an autoregressive manner. Despite being simple and effective, this method results in sub-optimal cross-modal alignment by over-emphasizing the text tokens that are less correlated with or even contradictory with the input images. In this paper, we advocate for assigning distinct contributions for each text token based on its visual correlation. Specifically, we present by contrasting image inputs, the difference in prediction logits on each text token provides strong guidance of visual correlation. We therefore introduce **C**ontrastive **AL**ignment (*CAL*), a simple yet effective re-weighting strategy that prioritizes visually correlated tokens. Our experimental results demonstrate that *CAL* consistently improves different types of VLMs across different resolutions and model sizes on various benchmarks. Importantly, our method incurs minimal additional computational overhead, rendering it highly efficient compared to alternative data scaling strategies.

## 1 Introduction

Recent advancements in Large Language Models (LLMs) [1–4] have opened up new avenues in multimodal understanding, giving rise to a novel model category known as Vision Language Models (VLMs) [5–8]. Many recent studies on VLMs are centered around enhancing their capabilities, either through increasing the resolution of input images [9–12] or incorporating higher-quality training datasets [13–16]. Additionally, research efforts have been directed towards exploring variations of vision models, such as replacing or augmenting vision encoders beyond CLIP [17, 18], including approaches like SigLIP [19], DINO [20, 21] or ConvNeXt [22, 23]. The integration of these techniques has spurred the development of VLMs, continually enhancing their performance across various benchmarks including visual question answering [24–28], image captioning [29, 30], and visual grounding [31, 32].

Despite these advancements, whether the current alignment strategy on existing image-text datasets performs satisfactorily is often less studied. Existing alignment strategies simply treat all text tokens equally in an auto-regressive manner. Although such a method has been proven to be simple and effective, many text tokens exhibit limited relevance to the visual inputs, which contribute little to image-text modality alignment. Figure 1a presents a sample drawn from the ShareGPT4V [16] dataset, where a large proportion of text tokens including *unique, context* presents little visual correlation. Treating these text tokens in equal weights results in ineffective training and can introduce negative effects by prioritizing more on fitting the distribution of these visually irrelevant tokens, rather than the image-text modality alignment.

---

[*]Equal contribution
[†]Work done during internship in ByteDance

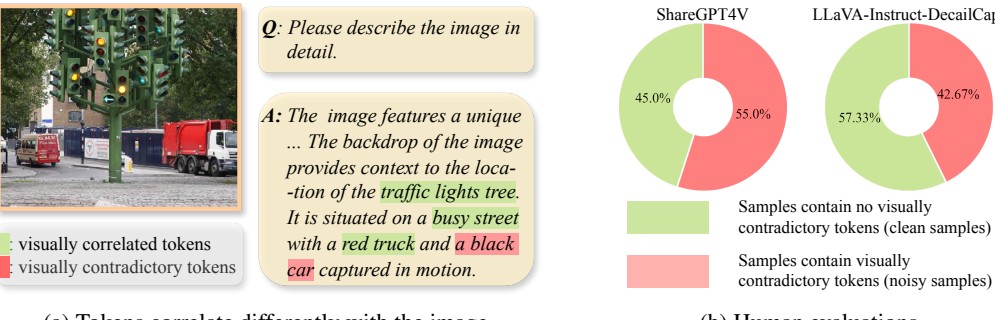

(a) Tokens correlate differently with the image.  (b) Human evaluations.

Figure 1: Figure 1a is one sample drawn from the ShareGPT4V dataset, which contains text tokens that are even contradictory with the given image. Figure 1b further presents our human evaluation results on the proportion of noisy samples that contain contradictory tokens.

Moreover, there also exists a proportion of tokens that contradicts visual conditions, which is inevitable in model generated datasets [7, 16, 13, 15, 14], presented in Figure 1a. In particular, we conduct human evaluations on the broadly used GPT-assisted datasets including ShareGPT4V and the detail caption subset of LLaVA-Instruct in Figure 1b by sampling 100 samples from each dataset. We score each sample by 0 and 1 based on whether there exist text tokens that contradict with visual input, and the score is averaged by three annotators. We found approximately half of the sampled datasets contain visually contradictory tokens in both datasets. Imitating the text distribution on these contradictory tokens further harms the image-text modality alignment. Consequently, recent evaluations on existing VLMs [33, 34, 5, 35, 6, 8] present the shortcomings of current alignment strategy from various aspects, including hallucination [36, 37, 33] and responding without depending on visual conditions [27, 38].

Fortunately, inspired by recent training-free visual contrastive decoding researches [36, 33, 39], we present that the visual correlation can be directly indicated by contrasting input image conditions. In particular, we investigate the change in prediction logits of text tokens with or without the image input and observe strong relevance between the logit change of each text token and its visual correlation. We therefore propose **C**ontrastive **AL**ignment (*CAL*), which is a surprisingly simple re-weighting strategy to prioritize the training of text tokens that are highly correlated with the input image to enhance image-text modality alignment. Experiments have shown that our proposed method can improve leading VLMs of different kinds including LLaVA-1.5/LLaVA-NeXT [7, 6, 10], MiniGemini(MGM)/MGM-HD [11], across different resolution and model size on various types of benchmarks including visual question answering, captioning and grounding. Especially, *CAL* on LLaVA-Next-13B [10] can bring an impressive performance of 1.7 ANLS on VQA$^{Doc}$ [26], 3.4 relaxed accuracy on VQA$^{Chart}$ [25], 2.2 CIDEr [40] on COCO [30] and 6.3 CIDEr on TextCaps [29], 0.6/0.7 IoU on validation/test set of RefCOCOg [32]. Moreover, our method introduces little computational overhead, with one auxiliary gradient-free forward operation in each training step. The lightweight feature while the impressive performance of *CAL* brings current VLMs to a new stage, highlighting the importance of a delicate image-text modality alignment strategy design. We further conduct extensive qualitative analysis for *CAL* and present the improved ability of OCR recognition and image-captioning.

In summary, our contributions are listed as follows.

- We present that by contrasting image inputs, existing VLMs are able to distinguish visually correlated tokens both visually irrelevant and visually contradictory ones.

- We propose *CAL* , a contrastive image-text alignment method via token re-weighting, which is lightweight and effective. *CAL* requires little additional training cost and no additional inference cost.

- Experiments show that our *CAL* can consistently improve VLMs of different kinds, across different resolutions and sizes in various types of benchmarks.

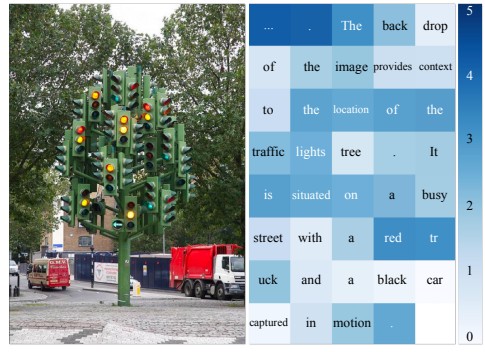

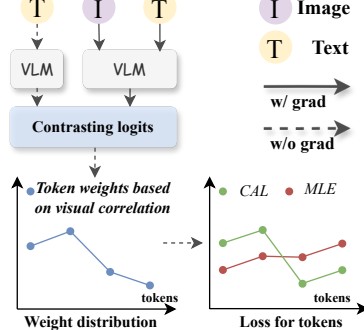

(a) $\Delta\mathbf{o}^{i,j}_{[t_j]}$ heat map.

(b) Training procedures of *CAL*

Figure 2: Overview of *CAL* . Figure 2a presents a sample drawn from the ShareGPT4V dataset. We calculate the logit difference w/ or w/o image inputs and plot the heat map on partial text tokens. Figure 2b presents the training procedure of *CAL* , which re-weights the importance of label tokens based on the contrasting logits.

## 2   Contrastive Alignment

In this section, we describe the detailed design of *CAL*. First of all, we review the existing image-text modality alignment method and provide the notations in Section 2.1. Secondly, we show the token discrepancy in cross-modal datasets can be inferred via contrasting image inputs in Section 2.2. Finally, we present a detailed description of our proposed *CAL* in Section 2.3.

### 2.1   Preliminary and Notations

**Preliminary**   Most existing VLMs adopt a two-stage strategy to align pre-trained image features with text embeddings in Large Language Models, i.e., a PreTraining (PT) stage that uses quantitative while noisy datasets for rough alignment, and an Instruction-Tuning (IT) stage that uses high-quality datasets to enhance the alignment. Both stages treat all tokens equally in an auto-regressive generation manner, i.e., the Maximum Likelihood Estimation (MLE) objective.

**Notations**   In this paper, we denote the alignment dataset $\mathbf{D}$ consisting paired image-text samples $\mathbf{D} = \{(I^1, T^1), (I^2, T^2), ..., (I^n, T^n)\}$, and denote the logit computation function as $f(\theta)$ where $\theta$ is the weight of VLMs. For the $i^{th}$ sample $(I^i, T^i)$ in the training dataset, where $T^i$ consists of a sequence of $l$ tokens $T^i = [t^{i,1}, t^{i,2}, ..., t^{i,l}]$, we denote the prediction logit distribution without input $I$ as $\mathbf{o}$, and prediction logit distribution with input $I$ as $\tilde{\mathbf{o}}$, depicted in the following equation:

$$\mathbf{o}^{i,j} = f_\theta(T^{i,<j}), \tilde{\mathbf{o}}^{i,j} = f_\theta(I^i, T^{i,<j}) \tag{1}$$

where $T^{i,<j}$ represents all previous tokens before position $j$ in the $i^{th}$ sample. We further use $\mathbf{o}^{i,j}_{[t_j]}$ or $\tilde{\mathbf{o}}^{i,j}_{[t_j]}$ to represent the prediction logit in the $i^{th}$ sample at token $t_j$. As a result, the MLE loss objective for the given $i^{th}$ sample at token $t_j$ is written in the following equation by treating the weight $c$ of each token equally, where $c$ is set to 1:

$$\mathcal{L}^{i,t_j}_{MLE} = c \cdot \log_{\text{softmax}}(\tilde{\mathbf{o}}^{i,j}_{[t_j]}) \tag{2}$$

### 2.2   Tokens Differ in Image-text Modality alignment

In this section, we present the necessity of token re-weighting in Section 2.2.1, and show that the re-weighting guidance could be naturally inferred by contrasting image inputs in Section 2.2.2.

#### 2.2.1   Discrepancy exists in text tokens

Our proposed method begins with the discrepancy in the training label tokens. For image-text modality alignment, the training labels are usually natural texts, where not all text tokens have a

strong correlation with the image inputs. Moreover, due to the existence of model generated datasets, there also exist noisy text tokens that harm the alignment process, which is depicted in Figure 1.

Based on the relevance between corresponding input images, text tokens can be naturally divided into three kinds. (1) **Visually correlated tokens**, which contain clear visual concepts and movements depicted in the image. (2) **Visually irrelevant tokens**, which contain either irrelevant to the image inputs or could be easily inferred by previous text tokens. (3) **Visually contradictory tokens**, which contain hallucinated objects, especially in the model generated datasets.

### 2.2.2 Visually correlation can be inferred by contrasting image inputs.

Inspired by VCD [36] and IBD [33], which both enhance the generation by contrasting image inputs, we further present that the contrastive method can also provide clear guidance for visually correlation on each token.

We take the prediction logit of each label token under two circumstances, i.e., with or without the image inputs, which we denote as $\tilde{\mathbf{o}}_{[t_j]}^{i,j}$ and $\mathbf{o}_{[t_j]}^{i,j}$. Then denote $\Delta\mathbf{o}_{[t_j]}^{i,j} = \tilde{\mathbf{o}}_{[t_j]}^{i,j} - \mathbf{o}_{[t_j]}^{i,j}$ as the difference between the predictions logits across two circumstances, we plot $\Delta\mathbf{o}_{[t_j]}^{i,j}$ on each token in Figure 2a to visualize the effect of image conditions.

From Figure 2a, $\Delta\mathbf{o}_{[t_j]}^{i,j}$ performs impressively in distinguishing text tokens of three kinds. By $\Delta\mathbf{o}_{[t_j]}^{i,j}$, the visually correlated tokens *the traffic lights tree, busy street, red truck* are specially high-lighted while other tokens, especially the visually contradictory tokens *a black car* are light-colored.

In summary, the discrepancy in the label tokens motivates us to apply the token-wise dynamics on loss to enhance the image-text modality alignment. By contrasting the image inputs, the difference in the prediction logits $\Delta\mathbf{o}_{[t_j]}^{i,j}$ aids us with clear guidance for visually correlation of each text token.

### 2.3 Contrastive Alignment (*CAL*)

In this section, we present the details of our proposed *CAL*. *CAL* proposed to re-assign the contribution of each token based on their visually correlation weights. The overview of our method is shown in Figure 2b and the detailed algorithm is depicted in Algorithm 1.

Following the previous section, *CAL* first dynamically computes the visually correlation weight $\Delta\mathbf{o}_{[t_j]}^{i,j}$ of each token $t_j$ by contrasting the image conditions. To avoid the effects of extreme values, we additionally introduce post-processing methods including clamping and average pooling. We clamp $\Delta logit$ by setting the upper bound to $\beta$ and the lower bound to $\alpha(\alpha >= 0)$. By setting $\alpha$ to the extreme value 0, *CAL* neglects the visually irrelevant tokens and visually contradictory tokens. By setting $\beta$ to the extreme value $+\infty$, *CAL* tolerates the circumstances where some visually correlated tokens occupy most of the importance weights in all label tokens:

---
**Algorithm 1** Detail Procedure of $\mathcal{L}_{CAL}^{i,t_j}$

**Input:** $i^{th}$ Image $I^i$, $i^{th}$ Text $T = t_1, t_2, ..., t_n$, VLM $f_\theta$

1: Compute contrastive logit.
    $\mathbf{o}^{i,j} = f_\theta(T^{i,<j}), \tilde{\mathbf{o}}^{i,j} = f_\theta(I^i, T^{i,<j})$
2: Compute $\Delta logit$.
    $\Delta\mathbf{o}_{[t_j]}^{i,j} = \tilde{\mathbf{o}}_{[t_j]}^{i,j} - \mathbf{o}_{[t_j]}^{i,j}$
3: Compute weights by post-processing $\Delta\mathbf{o}_{[t_j]}^{i,j}$.
    $\tilde{\mathbf{w}}^{i,t_j} = pooling_W(clamp_{\alpha,\beta}(\Delta\mathbf{o}_{[t_j]}^{i,j}))$
4: Compute CAL loss by re-weighting tokens.
    $\mathcal{L}_{CAL}^{i,t_j} = -\frac{1}{\sum_{k=1}^{l} \tilde{\mathbf{w}}^{i,t_k}} \tilde{\mathbf{w}}^{i,t_j} \cdot \log_{\text{softmax}} f_\theta(\tilde{\mathbf{o}}_{[t_j]}^{i,j})$

**Output:** $\mathcal{L}_{CAL}^{i,t_j}$

---

$$\mathbf{w}^{i,t_j} = clamp_{\alpha,\beta}(\Delta\mathbf{o}_{[t_j]}^{i,j}) \tag{3}$$

We further introduce average pooling with a window size of $W$ to smooth the visually correlation weights of each token, where we denote as:

$$\tilde{\mathbf{w}}^{i,t_j} = pooling_W(\mathbf{w}^{i,t_j}) \tag{4}$$

The final loss objective of *CAL* is the weighted average of the original MLE objective based on $w$ and is defined as:

$$\mathcal{L}_{CAL}^{i,t_j} = -\frac{1}{\sum_{k=1}^{l} \tilde{\mathbf{w}}^{i,t_k}} \tilde{\mathbf{w}}^{i,t_j} \cdot \log_{\text{softmax}} f_\theta(\tilde{\mathbf{o}}_{[t_j]}^{i,j}) \tag{5}$$

Table 1: Visual Question Answering benchmarks of *CAL* on leading methods including LLaVA-1.5, LLaVA-NeXT[1], and MGM/MGM-HD. Our results are marked with ▨. VQA$^{\text{Text}}$ is evaluated without OCR tokens. Abbreviations: OCRB. (OCR-Bench), MMS. (MMStar), MMT. (MMT-Bench).

| Method | LLM | OCRB. | VQA | | | SQA$^I$ | MMS. | MMT. | Win/All |
|--------|-----|-------|-----|-----|-----|--------|------|------|---------|
| | | | Doc | Chart | Text | | | | |
| *Low resolution setting* | | | | | | | | | |
| MGM | Gemma-2B | 335 | 39.8 | 23.4 | 48.1 | 60.6 | **25.5** | 43.4 | 6 / 7 |
| MGM+*CAL* | Gemma-2B | **360** | **44.8** | **27.0** | **51.8** | **64.0** | 25.4 | **45.4** | |
| MGM | Vicuna-7B | 431 | 57.7 | **43.2** | 61.1 | 69.9 | 32.8 | 50.3 | 6 / 7 |
| MGM+*CAL* | Vicuna-7B | **443** | **58.0** | 42.8 | **63.0** | **70.4** | **35.5** | **51.4** | |
| MGM | Vicuna-13B | 452 | **61.7** | **48.8** | 62.6 | 69.1 | 30.4 | 49.1 | 5 / 7 |
| MGM+*CAL* | Vicuna-13B | **466** | 61.6 | 48.0 | **63.8** | **71.9** | **33.7** | **51.9** | |
| LLaVA-1.5 | Vicuna-7B | 315 | 28.5 | 17.5 | **47.6** | 68.2 | 32.4 | 48.6 | 5 / 7 |
| LLaVA-1.5+*CAL* | Vicuna-7B | **328** | **30.6** | 17.5 | 47.5 | **68.7** | **32.9** | **48.8** | |
| LLaVA-1.5 | Vicuna-13B | 341 | 31.1 | 18.3 | 49.0 | 72.1 | 33.5 | 51.1 | 7 / 7 |
| LLaVA-1.5+*CAL* | Vicuna-13B | **347** | **32.6** | **18.4** | **49.6** | **72.7** | **35.7** | **51.2** | |
| *High resolution setting* | | | | | | | | | |
| MGM-HD | Vicuna-7B | 477 | 72.0 | 49.3 | 65.5 | 68.4 | **31.0** | 47.9 | 6 / 7 |
| MGM-HD+*CAL* | Vicuna-7B | **503** | **73.4** | **49.6** | **67.1** | **69.2** | 30.1 | **50.5** | |
| MGM-HD | Vicuna-13B | 502 | 77.7 | 55.8 | 67.2 | **73.5** | 34.2 | 50.9 | 6 / 7 |
| MGM-HD+*CAL* | Vicuna-13B | **535** | **78.0** | **57.2** | **68.8** | 73.1 | **38.5** | **51.4** | |
| LLaVA-NeXT | Vicuna-7B | 542 | 75.1 | 62.2 | 64.2 | 68.5 | 33.7 | 49.5 | 7 / 7 |
| LLaVA-NeXT+*CAL* | Vicuna-7B | **561** | **77.3** | **64.3** | **65.0** | **70.1** | **35.5** | **50.7** | |
| LLaVA-NeXT | Vicuna-13B | 553 | 78.4 | 63.8 | 67.0 | **71.8** | 37.5 | 50.4 | 6 / 7 |
| LLaVA-NeXT+*CAL* | Vicuna-13B | **574** | **80.1** | **67.2** | **67.1** | 71.5 | **38.1** | **52.4** | |

## 3 Experiments

### 3.1 Experimental Setup

**Implementation Details** In this paper, we verify our proposed *CAL* on two leading model structures: LLaVA-1.5/LLaVA-NeXT [6, 10] and Mini-Gemini/Mini-Gemini-HD [11]. LLaVA-1.5 uses CLIP-pretrained ViT-L as the visual encoder. For resolution scaling, LLaVA-NeXT employs a simple while adaptive image cropping strategy, encodes each image and concatenates them in one single sequence. Mini-Gemini (MGM) further introduces a LAION-pretrained ConvNeXt-L [22, 41] for high-resolution refinement. For MGM/MGM-HD/LLaVA-1.5, we follow the same setting as the original paper as it is public available, where the learning rate for the PT stage is set to $1e^{-3}$ and the IT stage is set to $2e^{-5}$ for both Vicuna-7B and Vicuna-13B. For LLaVA-NeXT, where only the evaluation code is made public, we reproduce LLaVA-NeXT with the same learning rate as MGM, and set the learning rate of ViT to 1/10 of the base learning rate (our reproduction presents on-par performance with the original paper/blog. We present a comparison of our reproduction results with those of the original papers in Appendix A.1). We also set the lower bound $\alpha$ and upper bound $\beta$ in Equation (3) to 1 and 5 respectively, and we set $l$ in Equation (4) to 3 for all experiments. We use 16 A100 for experiments, except for 8 GPUs in LLavA-1.5/Gemma-2B and 32 GPUs in MGM-HD-13B.

**Datasets** For experiments on LLaVA-NeXT [10], since the detailed composition of training datasets is not publicly available, we use a slightly different training dataset combination, where we include the mixture of LLaVA$_{665k}$ [6], VQA$^{\text{Doc}}$ [26], VQA$^{\text{Chart}}$ [25] and the ShareGPT4V [16]. For experiments of LLaVA-1.5 [6] and MGM/MGM-HD [11], we use the same dataset combination with original paper. The training datasets include LLaVA-filtered CC3M [42], ALLaVA [14], ShareGPT4V [16], LAION-GPT-4V [43], LIMA [44], OpenAssistant2 [45], VQA$^{\text{Doc}}$ [26], VQA$^{\text{Chart}}$ [25], DVQA [46] and AI2D [47]. Finally, we report results on widely-adopted VLM benchmarks, including VQA$^{\text{Text}}$ [48](without providing OCR tokens), VQA$^{\text{Doc}}$ [26], VQA$^{\text{Chart}}$ [25], OCR-Bench [49], MMT [28], MMStar [27], SQA$^I$ [50], COCO Caption [30], TextCaps [29], and RefCOCOg [32] in our main experiments which observe significant improvement in majority settings, and additional benchmarks MME [24], POPE [37], SEED-I [51], VQA$^{\text{Text*}}$ [48](with OCR tokens given), with comparable performance in Appendix A.2.

---

[1]We reproduce the results on LLaVA-NeXT on a slightly different training data combination, since the instruction-tuning data is not made public.

Table 2: Image captioning and visual grounding benchmarks on LLaVA-1.5, LLaVA-NeXT, and MGM/MGM-HD[2]. Our results are marked with ▨.

| Method | LLM | COCO Caption | TextCaps | Refcocog$_{val}$ | Refcocog$_{test}$ | Win/All |
|---|---|---|---|---|---|---|
| *Low resolution setting* | | | | | | |
| MGM | Gemma-2B | 8.0 | 14.1 | 46.2 | 46.7 | 4 / 4 |
| MGM+*CAL* | Gemma-2B | **29.7** | **25.1** | **50.9** | **51.1** | |
| MGM | Vicuna-7B | 18.2 | 31.4 | 66.4 | 66.7 | 4 / 4 |
| MGM+*CAL* | Vicuna-7B | **21.8** | **33.6** | **68.5** | **68.6** | |
| MGM | Vicuna-13B | 17.6 | 27.1 | 73.2 | 73.5 | 4 / 4 |
| MGM+*CAL* | Vicuna-13B | **18.6** | **32.1** | **73.7** | **74.1** | |
| LLaVA-1.5 | Vicuna-7B | 111.1 | 100.7 | 72.2 | 72.2 | 4 / 4 |
| LLaVA-1.5+*CAL* | Vicuna-7B | **111.8** | **106.7** | **73.5** | **72.6** | |
| LLaVA-1.5 | Vicuna-13B | 115.9 | 102.4 | 74.6 | 75.2 | 3 / 4 |
| LLaVA-1.5+*CAL* | Vicuna-13B | **119.4** | **105.6** | **75.3** | 75.2 | |
| *High resolution setting* | | | | | | |
| MGM-HD | Vicuna-7B | **33.4** | 42.4 | 70.2 | 70.6 | 3 / 4 |
| MGM-HD+*CAL* | Vicuna-7B | 31.3 | **52.6** | **70.9** | **71.5** | |
| MGM-HD | Vicuna-13B | 22.2 | 42.1 | 76.8 | **77.7** | 3 / 4 |
| MGM-HD+*CAL* | Vicuna-13B | **30.0** | **51.6** | **77.2** | 77.1 | |
| LLaVA-NeXT | Vicuna-7B | 112.0 | 115.0 | 77.6 | 77.5 | 4 / 4 |
| LLaVA-NeXT+*CAL* | Vicuna-7B | **114.7** | **124.7** | **78.4** | **78.1** | |
| LLaVA-NeXT | Vicuna-13B | 118.5 | 118.2 | 79.8 | 79.6 | 4 / 4 |
| LLaVA-NeXT+*CAL* | Vicuna-13B | **120.6** | **124.4** | **80.4** | **80.3** | |

## 3.2 Main Results

***CAL* effectively improves various VLMs in Visual Question Answering scenarios** Table 1 presents the performance of VLMs on Visual Question Answering. Our *CAL* consistently improves the performance on most of the understanding benchmarks with impressive margin, on different resolution settings on both MGM/MGM-HD and LLaVA-1.5/LLaVA-NeXT which have different vision model architectures. Especially on the high resolution setting, our *CAL* presents impressive performance improvement on OCR centric benchmarks. *CAL* can bring improvement in 7 out of 8 benchmarks on LLaVA-NeXT-7B, MGM-HD-7B/13B, and 6 out of 8 benchmarks on LLaVA-NeXT-13B. Especially, on LLaVA-NeXT-13B, *CAL* improves VQA$^{Doc}$ by 1.7 ANLS, VQA$^{Chart}$ by 3.4 relaxed accuracy, and OCR-Bench by 21 points.

**Effectiveness of *CAL* is consistent on other benchmarks including Image captioning and Grounding** More promisingly, we found the improvement of *CAL* is not limited to Visual Question Answering benchmarks, but even more impressive on image-caption and visual grounding benchmarks. Table 2 presents the performance of VLMs on COCO Caption, TextCaps and both validation and test set of RefCOCOg. Especially, *CAL* improves LLaVA-NeXT-13B 2.1 CIDEr score on COCO Caption benchmark, 6.2 CIDEr score on TextCaps, 0.6/0.7 IoU on the validation/test set of RefCOCOg.

## 3.3 Ablation Studies

**Stages of Conducting *CAL* within VLM Training** *CAL* can be integrated into both the PreTraining (PT) stage and the Instruction Tuning (IT) stage in existing VLMs. In this section, we investigate which stage benefits the most from *CAL* in Table 3. Integrating *CAL* into the IT stage contributes most of the improvement in all listed benchmarks, and *CAL* in the PT stage further enhances the performance with notable improvement on MMT-Bench and OCR-Bench.

***CAL* relieve the effect of noisy labels in the training data** *CAL* can impressively reduce the effect of the contradictory label tokens in training data. To distinctly presents the denoising ability of *CAL*, we pollute the original training data with a ratio of 10 %, 20 % and 30 % by exchanging the training images and their labels, i.e., we select $(I^i, T^i)$ and $(I^j, T^j)$ with a probability of $p$ and replace them with $(I^i, T^j)$ and $(I^j, T^i)$. We plot the performance difference when the noise rate is varied on four

---

[2]Note that the caption score of MGM/MGM-HD is significantly lower than LLaVA-NeXT. The training split of MGM does not include the training set for both COCO Caption and TextCaps, where we provide the zero-shot performance for MGM in all settings on these two benchmarks.

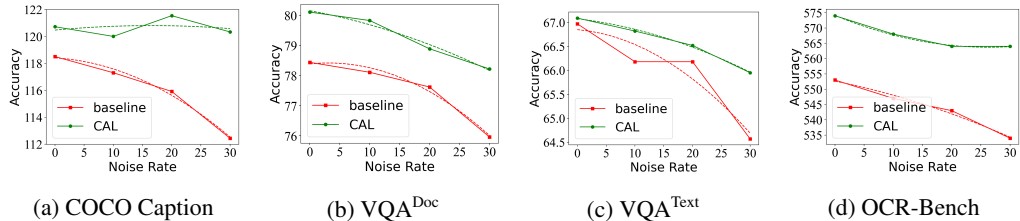

| (a) COCO Caption | (b) VQA$^{\text{Doc}}$ | (c) VQA$^{\text{Text}}$ | (d) OCR-Bench |

Figure 3: Accuracy difference when different noise ratios applied. The performance of the baseline is marked with red lines, and *CAL* is marked with green lines. The dashed line represents the asymptote.

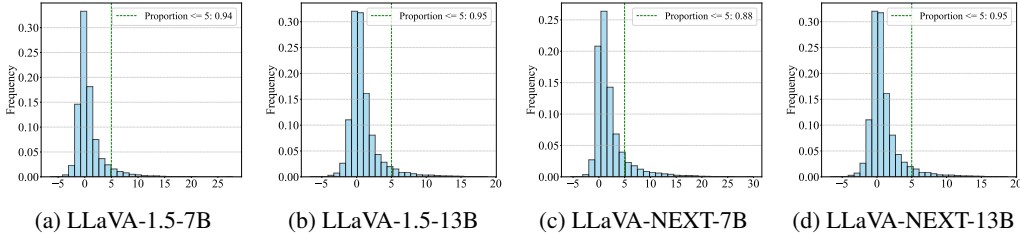

| (a) LLaVA-1.5-7B | (b) LLaVA-1.5-13B | (c) LLaVA-NEXT-7B | (d) LLaVA-NEXT-13B |

Figure 4: $\Delta\mathbf{o}$ distribution for LLaVA models on 100 random sampled cases.

Table 3: Performance difference when *CAL* is applied at different training stages.

| PT | IT | VQA$^{\text{Doc}}$ | TextCaps | MMT. | OCRB. |
|----|----|------|----------|------|-------|
| | | 78.4 | 118.1 | 50.4 | 553 |
| ✓ | | 78.5 | 116.7 | 50.9 | 554 |
| | ✓ | 79.9 | 124.8 | 51.5 | 564 |
| ✓ | ✓ | 80.1 | 124.4 | 52.4 | 574 |

Table 4: Performance difference when applying different weights $[\alpha, \beta]$ for clamping.

| $[\alpha, \beta]$ | VQA$^{\text{Doc}}$ | VQA$^{\text{Chart}}$ | OCRB. | Refcocog$_{\text{val}}$ |
|-------------------|------|------|------|--------|
| $[0, +\infty]$ | 79.4 | 66.6 | 563 | 79.4 |
| $[0, 5]$ | 80.4 | 66.3 | 570 | 79.7 |
| $[1, +\infty]$ | 79.1 | 66.8 | 571 | 78.5 |
| $[1, 5]$ | 80.1 | 67.2 | 574 | 80.4 |
| Baseline | 78.4 | 63.8 | 553 | 79.8 |

different benchmarks, including COCO caption, VQA$^{\text{Doc}}$, VQA$^{\text{Text}}$ and OCR-Bench. From Figure 3, the performance of baseline drops significantly when the noise rate is increased, while *CAL* performs more robust to noise.

**Hyper-parameters for $[\alpha, \beta]$ in clamping** We further conduct ablation study on $\alpha$ and $\beta$ to study the effect of the hyperparameters in our clamping operation.

First, we plot the $\Delta\mathbf{o}$ distribution on LLaVA series in Figure 4. Tokens whose $\Delta\mathbf{o}$ is lower than 5 nearly occupy approximately 90% of the total label sequences. Therefore, we select 5 as the upper bound $\beta$. To prevent the context dependent tokens being totally neglected, we set the lower bound $\alpha$ to 1. We then extend both lower bound and upper bound to extreme values, i.e., 0 and $+\infty$. The results are shown in Table 4. (1) Both setting the lower bound to 0 and setting the upper bound to $+\infty$ causes performance degradation. (2) Even when setting the lower bound to 0, *CAL* still achieves better performance compared with the original baseline, indicating that totally neglecting the context dependent/visually contradictory tokens while focusing on the visually dependent ones still brings steady improvement.

**Complementary Ablations** We further provide ablations on the image contrasting conditions in Appendix A.3, where we compare *CAL* when different kinds of augmentation strategy is used.

### 3.4 Analysis

***CAL* presents better capability in OCR recognition and capturing details** We provide qualitative analysis in Table 5 to analyze the improved caption ability of *CAL*. In the table, *CAL* presents much better ability in OCR recognition by accurately distinguishing bewildering OCR cases, e.g., *CAL* accurately tells the difference between hand-written *consciousness* and *construction*, *world* and *would*. Such ability contributes to the superior performance of *CAL* on the TextCaps benchmark.

Meanwhile, *CAL* also presents better ability in capturing visually-conditioned details. For instance, compared with baseline, *CAL* captures the material details *cd cover*, and the numerical details by telling *two men on horses* from *a man on a horse*. The capability of *CAL* to capture intricate details leads to sustained enhancements in the COCO caption benchmark. *CAL* empowers the model to identify more accurate elements within images, including objects like *a trolley* and *the number 8*, which might otherwise be incorrectly recognized or overlooked.

**Complementary Analysis**  We first provide statistics for computational overhead in Appendix A.6. And we further provide more qualitative analysis on studying the quality of image-text modality alignment. The attention map scores are visualized in Appendix B.1, and the aligned image features are visualized in Appendix B.2.

## 4    Related Work

**Vision Language Models**  LLMs [1, 52, 2, 53, 4] have made significant strides in Natural Language Processing (NLP) tasks, including text generation and question-answering, paving the way for VLMs that integrate vision ability with LLMs. In the realm of visual language learning, CLIP [17, 54] has set a milestone by employing extensive image-text pair contrastive learning to achieve multimodal alignment. Recently, numerous VLMs [5, 7, 8, 55, 34, 56–59, 35, 60, 61] have leveraged the robust capabilities of LLMs for cross-modal understanding and generation tasks. Models like BLIP-2 [5] and MiniGPT-4 [8] have improved cross-modal alignment through comprehensive image-text pair pre-training. LLaVA [7] has further advanced its comprehension of complex prompts via refined instruction fine-tuning. Additionally, recent research [9–11] has incorporated higher resolution input images and longer sequences to enhance VLMs' understanding capabilities. Mini-Gemini (MGM) [11] introduces a LAION-pretrained ConvNeXt-L [22] for high-resolution refinement.

**Image-text Modality Alignment**  Image-text modality alignment has long been regarded as the core problem in cross-modal understanding and generation tasks. Traditional image-text alignment strategies include both contrastive learning across different modalities and generative learning that train text tokens in an autoregressive manner [17, 62–64]. The combination of both techniques is also proven to be effective in the early era of VLMs, where BLIP [5] proposes a multi-stage alignment strategy, with contrastive learning in early alignment and generative learning in the latter stage. However, in recent researches [7, 10], contrastive learning is discarded for being redundant in image-text modality alignment of VLMs, and researchers propose to enhance the cross-modal alignment through dataset scaling and image resolution scaling [53, 9, 65, 10]. Despite being simple in application, the existing generative alignment method simply treats each text token with equal importance, resulting in sub-optimal alignment performance.

More recently, due to the great success in Reinforcement Learning (RL) in the alignment of LLMs, many recent works [66, 37, 67] have also integrated Reinforcement Learning methods to align existing VLMs with human preference. However, these RL-based methods require high-quality human-labeled pair-wise data and focus more on aligning with human preference rather than modality alignment.

**Training-free Contrastive Decoding**  Recently, many researchers have proposed to improve the generation quality via contrastive decoding [68–70, 36, 33]. Especially, in the field of LLMs, CID [71] utilizes contrastive decoding on paired text inputs for model de-biasing. Such method is also proven to be effective in enhancing the reasoning ability of LLMs in various aspects [72–74]. Similar investigations have also been taking in VLMs. Recently, both VCD [36] and IBD [33] propose to enhance the generation of VLMs by contrasting the prediction logits between the original visual input and the perturbed ones. CRG [39] further proposes to improve the grounding ability without training via contrasting differently masked images. However, these training-free methods require additional computation during the decoding stage, making it highly ineffective for application.

## 5    Limitation

Despite the superiority of *CAL* in various model structures, resolution settings and model scales on various benchmarks, limitations still exist in our proposed method. First of all, there lacks a clear and

Table 5: OCR and captions generation comparison based on LLaVA-NEXT-13B model.

**OCR cases**

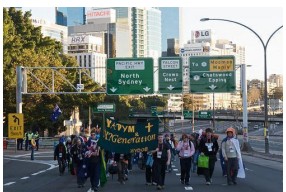 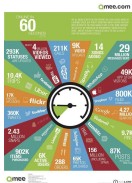

| | | |
|---|---|---|
| *Question:* | What is written in the image? | What is the number in the image? |
| Baseline: | The word "consciousness" is written in the image. | The word "world" is written in the image. |
| *CAL :* | The word "construction" is written in the image. | The word "would" is written in the image. |

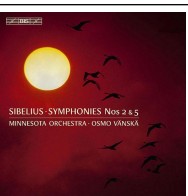 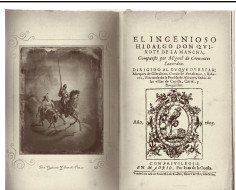

| | | |
|---|---|---|
| *Question:* | What is the Mosman Manly exit going to? | How many items purchased from Amazon? |
| Baseline: | The mosman manly exit is going to mosman. | There are 2.43 million items purchased from amazon. |
| *CAL :* | The mosman manly exit is going to chatswood epping. | 902k. |

**Short caption cases**

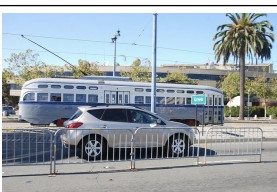 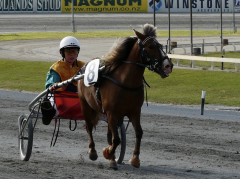

| | | |
|---|---|---|
| *Question:* | Provide a one-sentence caption for the provided image. | Provide a one-sentence caption for the provided image. |
| Baseline: | A red background with a sun and birds and the words Sibelius Symphonies No 2 & 5. | A book is open to a page with a picture of a man on a horse. |
| *CAL :* | A cd cover for Sibelius Symphonies Nos 2 & 5. | A book is open to a page with a picture of two men on horses. |

| | | |
|---|---|---|
| *Question:* | Provide a one-sentence caption for the provided image. | Provide a one-sentence caption for the provided image. |
| Baseline: | A silver car is parked in front of a fence and a bus. | A person riding a horse with a cart attached to it. |
| *CAL :* | A silver car is parked behind a fence in front of a trolley. | A horse pulling a cart with the number 8 on it. |

quantitative discrepancy between the three kinds of label tokens. More discussion of the importance weights guidance can be further investigated in future works.

The selection of lower bounds and upper bounds in Equation (3) are empirically decided based on the frequency of the prediction logits, which could be extended to more adaptive settings in further explorations. Nevertheless, the simple while broadly effective nature of *CAL* indicates the importance of a delicate image-text modality alignment strategy for leading VLM structures.

## 6   Conclusion

In this paper, we investigate the in-completeness of current image-text alignment in leading VLMs by treating all text tokens with equal weights. We present by contrasting input images, the difference in the prediction logits for each token naturally reveals their visual correlation. We therefore propose a token re-weighting strategy that prioritize the training of highly visually correlated tokens. Our proposed strategy, *CAL* is simple while impressively effective, achieving consistent performance gain across various benchmarks including visual question answering, image-captioning and grounding.

Our work raises a question about the potential optimal learning strategy of image-text modality alignment. Both the imperfectness of training data and over concentration on visually irrelevant/visually contradictory tokens hinder the performance of current VLMs. We hope the proposed *CAL* can inspire more investigation on better alignment strategy to enhance the capabilities of existing VLMs.

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

## Outline

# A Complementary Experimental Analysis

## A.1 Ablations for Comparison with Official Results

Table 6: Comparisons between our reproduced results and official results. ∗ denotes providing OCR tokens for evaluation. Official results are marked with †.

| Method | LLM | VQA | | | | SQA$^I$ | MME | POPE | SEED-I | COCO Caption | TextCaps |
|---|---|---|---|---|---|---|---|---|---|---|---|
| | | Doc | Chart | Text | Text* | | | | | | |
| *Low resolution setting* | | | | | | | | | | | |
| **MGM** | Gemma-2B | 39.8 | 23.4 | 48.1 | 56.2 | 60.6 | 1335 | 85.7 | 64.6 | 8.0 | 14.1 |
| **MGM**† | Gemma-2B | - | - | - | 56.2 | - | 1341 | - | - | - | - |
| **MGM** | Vicuna-7B | 57.7 | 43.2 | 61.1 | 65.7 | 69.9 | 1551 | 85.7 | 68.2 | 18.2 | 31.4 |
| **MGM**† | Vicuna-7B | - | - | - | 65.2 | - | 1523 | - | - | - | - |
| **MGM** | Vicuna-13B | 61.7 | 48.8 | 62.6 | 66.1 | 69.1 | 1550 | 86.2 | 69.5 | 17.6 | 27.1 |
| **MGM**† | Vicuna-13B | - | - | - | 65.9 | - | 1565 | - | - | - | - |
| **LLaVA-1.5** | Vicuna-7B | 28.5 | 17.5 | 47.6 | 58.2 | 68.2 | 1468 | 86.2 | 66.6 | 111.1 | 100.7 |
| **LLaVA-1.5**† | Vicuna-7B | 28.1 | 18.2 | 46.1 | - | 69.5 | 1508 | 85.9 | 66.2 | 110.4 | 98.1 |
| **LLaVA1.5** | Vicuna-13B | 31.1 | 18.3 | 49.0 | 60.6 | 72.1 | 1574 | 85.7 | 68.6 | 115.9 | 102.4 |
| **LLaVa1.5**† | Vicuna-13B | 30.3 | 18.2 | 48.7 | - | 72.9 | 1523 | 85.9 | 68.2 | 115.5 | 104.0 |
| *High resolution setting* | | | | | | | | | | | |
| **MGM-HD** | Vicuna-7B | 72.0 | 49.3 | 65.5 | 68.2 | 68.4 | 1521 | 85.7 | 65.7 | 33.4 | 42.4 |
| **MGM-HD**† | Vicuna-7B | - | - | - | 68.4 | - | 1546 | - | - | - | - |
| **MGM-HD** | Vicuna-13B | 77.7 | 55.8 | 67.2 | 69.8 | 73.5 | 1633 | 86.3 | 70.1 | 22.2 | 42.1 |
| **MGM-HD**† | Vicuna-13B | - | - | - | 70.2 | - | 1597 | - | - | - | - |
| **LLaVA-NeXT** | Vicuna-7B | 75.1 | 62.2 | 64.2 | 67.1 | 68.5 | 1514 | 86.8 | 69.6 | 112.0 | 115.0 |
| **LLaVA-NeXT**† | Vicuna-7B | 74.4 | 54.8 | 64.9 | - | 70.2 | 1519 | 86.4 | 70.2 | 99.9 | 71.8 |
| **LLaVA-NeXT** | Vicuna-13B | 78.4 | 63.8 | 67.0 | 70.0 | 71.8 | 1574 | 87.2 | 71.8 | 118.5 | 118.2 |
| **LLaVA-NeXT**† | Vicuna-13B | 77.5 | 62.2 | 66.9 | - | 73.6 | 1575 | 86.3 | 71.9 | 102.0 | 67.4 |

We compare our reproduced results with those reported in the paper or from reliable sources in Table 6. The results for Mini-Gemini are taken from their paper, while the results for LLaVA are sourced from this sheet provided by the LLaVA authors. We can observe that our reproduced results are comparable to the official ones.

## A.2 Results on other Visual Question Answering benchmarks.

Table 7: Results on additional Visual Question Answering benchmarks. ∗ denotes providing OCR tokens for VQA$^{\text{Text}}$. Our results are marked with ▨.

| Method | LLM | MME | POPE | SEED-I | VQA$^{\text{Text}*}$ |
|---|---|---|---|---|---|
| *Low resolution setting* | | | | | |
| **MGM** | Gemma-2B | **1335** | 85.7 | 64.6 | 56.2 |
| **MGM**+*CAL* | Gemma-2B | 1333 | **85.8** | **65.0** | **58.1** |
| **MGM** | Vicuna-7B | **1551** | 85.7 | 68.2 | 65.7 |
| **MGM**+*CAL* | Vicuna-7B | 1509 | **87.5** | **69.0** | **66.7** |
| **MGM** | Vicuna-13B | **1550** | 86.2 | **69.5** | 66.1 |
| **MGM**+*CAL* | Vicuna-13B | 1506 | **86.4** | 69.4 | **67.1** |
| **LLaVA-1.5** | Vicuna-7B | 1468 | **86.2** | **66.6** | 58.2 |
| **LLaVA-1.5**+*CAL* | Vicuna-7B | **1500** | 85.5 | 66.5 | **58.6** |
| **LLaVA1.5** | Vicuna-13B | **1574** | 85.7 | 68.6 | **60.6** |
| **LLaVa1.5**+*CAL* | Vicuna-13B | 1567 | **85.8** | **69.3** | 60.1 |
| *High resolution setting* | | | | | |
| **MGM-HD** | Vicuna-7B | 1521 | 85.7 | 65.7 | 68.2 |
| **MGM-HD**+*CAL* | Vicuna-7B | **1531** | **87.0** | **68.6** | **69.5** |
| **MGM-HD** | Vicuna-13B | **1633** | 86.3 | 70.1 | 69.8 |
| **MGM-HD**+*CAL* | Vicuna-13B | 1582 | **86.4** | **70.5** | **70.3** |
| **LLaVA-NeXT** | Vicuna-7B | **1514** | 86.8 | 69.6 | 67.1 |
| **LLaVA-NeXT**+*CAL* | Vicuna-7B | 1490 | 86.7 | **70.7** | 67.1 |
| **LLaVA-NeXT** | Vicuna-13B | 1574 | 87.2 | **71.8** | **70.0** |
| **LLaVA-NeXT**+*CAL* | Vicuna-13B | **1606** | 87.2 | 71.6 | 68.9 |

Table 8: Ablations for contrasting image conditions on Visual Question Answering benchmarks using LLaVA-NeXT/13B.

| Method | Mask | $\sigma$ | MMS. | MMT. | SQA$^I$ | VQA | | | | OCRB. |
| | | | | | | Text | Text* | Doc | Chart | |
|---|---|---|---|---|---|---|---|---|---|---|
| LLaVA-NeXT | - | - | 37.5 | 50.4 | 71.8 | 67.0 | 70.0 | 78.4 | 63.8 | 553 |
| LLaVA-NeXT+*CAL* | - | - | 38.1 | **52.4** | 71.5 | 67.1 | 68.9 | **80.1** | 67.2 | 574 |
| LLaVA-NeXT+*CAL$_{mask}$* | 0.5 | - | 37.9 | 51.9 | 70.5 | 67.1 | 69.5 | 79.0 | 66.0 | 555 |
| LLaVA-NeXT+*CAL$_{mask}$* | 0.7 | - | **38.9** | 51.3 | 72.1 | 66.1 | 69.2 | 79.3 | 65.5 | 541 |
| LLaVA-NeXT+*CAL$_{mask}$* | 0.9 | - | 38.3 | 52.0 | 71.8 | 67.0 | 69.4 | 78.9 | 65.9 | **580** |
| LLaVA-NeXT+*CAL$_{gaussian}$* | - | 1 | 37.9 | 51.7 | 71.2 | 66.8 | 69.6 | 78.7 | 65.7 | 554 |
| LLaVA-NeXT+*CAL$_{gaussian}$* | - | 10 | 38.5 | 52.0 | 70.2 | **67.3** | **70.2** | 79.1 | 64.6 | 564 |

Table 9: Ablations for image contrasting conditions on image captioning and visual grounding benchmarks using LLaVA-NeXT/13B.

| Method | Mask | $\sigma$ | COCO Caption | TextCaps | Refcocog$_{val}$ | Refcocog$_{test}$ |
|---|---|---|---|---|---|---|
| LLaVA-NeXT | - | - | 118.5 | 118.2 | 79.8 | 79.6 |
| LLaVA-NeXT+*CAL* | - | - | **120.6** | **124.4** | 80.4 | 80.3 |
| LLaVA-NeXT+*CAL$_{mask}$* | 0.5 | - | 114.6 | 122.9 | **80.9** | 80.8 |
| LLaVA-NeXT+*CAL$_{mask}$* | 0.7 | - | 116.0 | 122.9 | 80.6 | **81.7** |
| LLaVA-NeXT+*CAL$_{mask}$* | 0.9 | - | 117.7 | 121.5 | 80.6 | 81.1 |
| LLaVA-NeXT+*CAL$_{gaussian}$* | - | 1 | 117.3 | 118.9 | 80.3 | 80.1 |
| LLaVA-NeXT+*CAL$_{gaussian}$* | - | 10 | 116.8 | 123.4 | 79.2 | 79.8 |

In the primary sections of this manuscript, *CAL* achieves better performance on benchmarks of OCR centric VQA, image-captioning, visual grounding, and other image-dependent benchmarks. In this section, we extend our analysis by presenting additional results on MME, POPE and SEED-I and VQA$^{Text}$ (with OCR token given). *CAL* presents comparable performance on these benchmarks, without a clear distinction on performance in different training settings. With steady improvement on OCR centric or image dependent VQA benchmarks, *CAL* also presents satisfying quality on the remaining VQA tasks.

## A.3 Ablations for Image Contrasting Conditions

In this section, we further conduct additional analysis on the image contrasting conditions. Except from masking the whole image token sequence, which is the method we use in our paper, we further study two kinds of contrasting conditions, including random patch masking and Gaussian blurring. For random patch masking, we cut images into $20 \times 10$ grids, with 10 pieces in the width and 20 pieces in the height. Therefore each images is cut into 200 pieces. We then randomly select 50 %, 70 % and 90 % of the patches to mask, and contrast them with the original image. For Gaussian blurring, we set the kernel $\sigma$ to 1 and 10 to simulate the extreme blurring circumstances. By using 1, we adopt light blurring which only brings significant effect on OCR centric images. By using 10, all images will be heavily affected.

Table 8 presents the results on Visual Question Answering benchmarks of different image contrasting conditions on LLaVA-NeXT-13B, and Table 9 further presents results on image-caption benchmarks and visual grounding benchmarks. From these two tables, either random patch masking or adding Gaussian blurring performs sub-optimal in different benchmarks. By *CAL* mask, except for the significant improvement on the test split of RefCOCOg, performance of *CAL* Gaussian drops significantly, especially on the OCR-centric benchmarks. By *CAL* Gaussian, although the performance on OCR centric benchmarks (VQA$^{Doc}$, VQA$^{Chart}$, OCR-Bench) is less affected, it performs less effective in nearly all benchmarks than *CAL* .

## A.4 Ablations for Average Pooling

We fix the window size (W) to 3 in Equation (4) to smooth the weight distribution. An additional experiment in Table 10, where we removed the average pooling step, shows slightly inferior performance, supporting our chosen.

Table 10: Comparison of benchmarks with and without Average Pooling.

| | ChartVQA | DocVQA | SQA$^I$ | COCO Caption | TextCaps | OCRB. | Refcocog$_{val}$ |
|---|---|---|---|---|---|---|---|
| w/ AvgPool | 67.2 | 80.1 | 71.5 | 120.6 | 124.4 | 574 | 80.4 |
| w/o AvgPool | 66.3 | 79.5 | 72.4 | 116.7 | 123.8 | 581 | 79.5 |

## A.5 Ablations for Pre-trained Model

Table 11: Comparison of pre-trained models for *CAL* on LLaVA-Next-13B.

| Model | Pretrain | ChartVQA | DocVQA | SQA$^I$ | COCO Caption | TextCaps | OCRB. | Refcocog$_{val}$ |
|---|---|---|---|---|---|---|---|---|
| Baseline | Original | 63.8 | 78.4 | 71.8 | 118.5 | 118.2 | 553 | 79.8 |
| CAL | Original | 67.2 | 80.1 | 71.5 | 120.6 | 124.4 | 574 | 80.4 |
| CAL | Baseline | 66.3 | 79.5 | 72.4 | 116.7 | 123.8 | 581 | 80.2 |

One assumption of *CAL* is that after training a simple projector only, the VLM is capable of distinguishing visually correlated tokens. In the first phase of VLM training, it is common practice to freeze the ViT and the LLM, and only train a projector to align their features. In the second phase, we finetune the model using high-quality data to enable it to answer image-related questions. To validate this assumption, we finetune *CAL* models using two types of pre-trained versions: one with the original pre-trained model (only train the projector) and one with the fully trained baseline model (after the finetuning phase). As shown in Table 11, we compare the performance of these two versions and found no significant differences in performance.

## A.6 Computational Overhead.

Table 12: Training time comparison with and without *CAL* in the instruction-tuning stage on LLaVA-NeXT.

| Pretrain | LLaVA-NeXT-7B | LLaVA-NeXT-13B |
|---|---|---|
| Baseline | 15.6$h$ | 27.4$h$ |
| Baseline + *CAL* | 19.1$h$ | 33.7$h$ |

Each iteration of our proposed method requires forwarding text tokens twice to effectively enhance image-text modality alignment. Table 12 presents the training time for the instruct-tuning stage on LLaVA-NeXT, with each experiment conducted on 16 A100 GPUs. *CAL* introduces approximately 20% computational overhead on both LLaVA-NeXT-7B and LLaVA-NeXT-13B. Our implementation currently uses an attention mask to neutralize the effect of the image tokens, meaning that the image tokens are still forwarded twice. This additional burden could be further reduced by completely removing image tokens.

# B  Complementary Qualitative Analysis

## B.1  Attention Map Visualization

In this section, we visualize the attention maps generated by both the baseline model and our proposed model. These visualizations provide insight into how each model focuses on different parts of the input image. The attention weights are calculated by accumulating the attention score between image tokens and text tokens across all layers. As shown in Figure 5, model w/ *CAL* provides clearer attention maps with less noisy points in the background area, indicating a more precise focus on relevant regions.

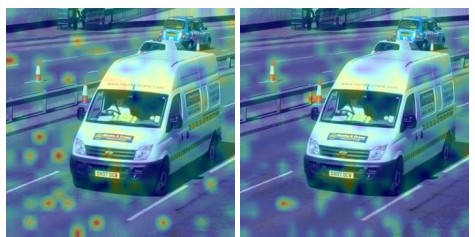
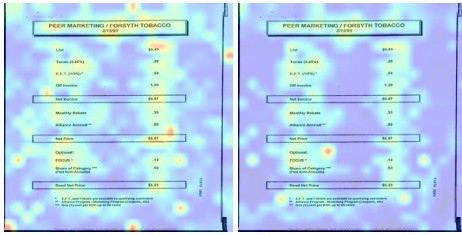

(a) a white van on a highway from Monks & Crane.      (b) $8.49.

Figure 5: Comparison of attention maps with and without *CAL* on LLaVA-NeXT-13B. The left side of each sub-figure shows LLaVA-NeXT-13B without *CAL* , while the right side shows LLaVA-NeXT-13B with *CAL* .

## B.2  Image-text Modality Alignment Visualization

In this section, we visualize the image-text modality alignment by retrieving the nearest text words to each image patch feature from the LLM vocabulary. We plot the results in Figure 6. *CAL* brings better image-text modality by accurately retrieving the OCR information from the language vocabulary. E.g., LLaVA1.5-7B + *CAL* correctly retrieves *Prices, expert, 0, shadow*, which is exactly the OCR information in the original image. We do not select LLaVA-NeXT in this section due to the presence of token overlaps from sub-crops.

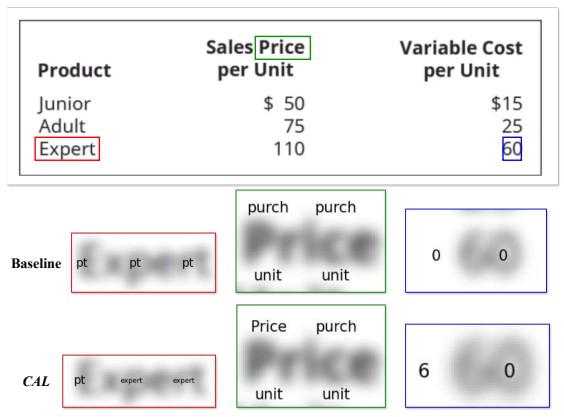
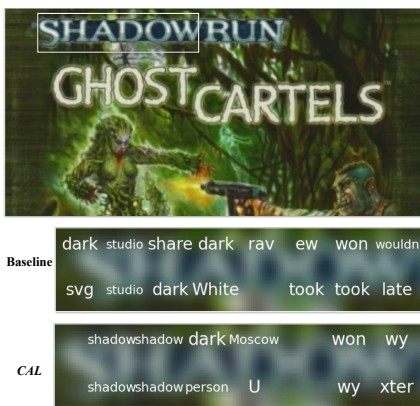

Figure 6: Visualization of image-text modality alignment for each image patch. We filtered out some nonsensical patches for better visualization.

