# OpenReview forum: "Seeing the Image: Prioritizing Visual Correlation by Contrastive Alignment"
_NeurIPS.cc/2024/Conference — NeurIPS 2024 poster_

### Official Review · Reviewer_mF2S · 2024-07-08

**Soundness:** 3
**Presentation:** 1
**Contribution:** 2
**Rating:** 5
**Confidence:** 4

**Summary:**

The paper proposes a new re-weighting strategy for training VLMs to improve visual-language alignment. The motivation is to assign higher weight to visually relevant tokens and lower weight to visually irrelevant and visually contradictory tokens. The re-weighting factor relies on the logit difference with and without image inputs for each token. Using the proposed CAL, the paper demonstrates general improvement in VQA and captioning tasks.

**Strengths:**

The motivation to dynamically assign weights to different types of tokens is intuitive and well-studied in the language domain. Several studies have attempted to apply this method to vision-language tasks in a zero-shot manner, which incurs high inference costs. This paper proposes training the model with this method instead. The approach is simple and effective, demonstrating general improvements over existing state-of-the-art models across benchmarks. The analysis further validates the effectiveness of the proposed method.

**Weaknesses:**

1. **Lack of Clarification in Presentation**: Some content is confusing. For example, equation 2 is averaged across the entire sentence with length $l$, yet the description in lines 88-89 refers to the loss objective for the $i^{th}$ sample at token $t_j$, which seems more like the loss at a single token. Additionally, equation 4 lacks clarification; it's unclear how to apply average pooling with a window size from this equation. Some experiments also lack necessary context. For instance, it's unclear which model is evaluated in Figure 2a, and the ablation study in section 3.3 doesn't specify which model is used for evaluation. The main result in section 3.3 doesn't disucuss how to apply CAL into model training, at which stage for example. The annotation is inconsistent; in equation 4, it's $\tilde{w}^{i,t_j}$, while in equation 5, it's $\tilde{w}(i.t_k)$

2. **More Experiments Are Required**: The paper presents decent improvements over some baselines, mainly in vision-language tasks. However, since the training recipe has changed, it would be better to also evaluate other tasks, such as language-only tasks, to see how the CAL affects the model's ability in standard language understanding. Additionally, the method seems quite relevant to hallucination tasks, so it would be beneficial to report results on those as well.

3. **Lack of Originality**: The motivation and methodology are already well-studied in language models [1,2], and prior studies have also applied changes to logits based on the presence or absence of images to measure the relevance of visual and language inputs. In this context, the paper simply applies the method during the training stage, instead of the inference stage as in [3]. Therefore, the novelty is somewhat limited.

[1] Jiang S, Wolf T, Monz C, et al. TLDR: token loss dynamic reweighting for reducing repetitive utterance generation[J]. arXiv preprint arXiv:2003.11963, 2020.

[2] Lin Z, Gou Z, Gong Y, et al. Rho-1: Not all tokens are what you need[J]. arXiv preprint arXiv:2404.07965, 2024.

[3] Zhu L, Ji D, Chen T, et al. Ibd: Alleviating hallucinations in large vision-language models via image-biased decoding[J]. arXiv preprint arXiv:2402.18476, 2024.

**Questions:**

1. The trend shown in Figure 2a is based on which model? Is this a common phenomenon across existing VLMs?

2. How do you deal with cases where a semantic word has been split into two subparts, each with a different value, such as "tr" and "uck" in Figure 2a?

3. The CAL seems to rely heavily on the model's logits with and without image input. However, can the model accurately predict changes in logits, as shown in Section 2, during the initial alignment stages?

4. The training requires an additional forward pass of the LLM. Why is the training described as "lightweight" in the paper? Doesn't this double the cost?

**Limitations:**

not applied

---

> ### Author Rebuttal · Authors · 2024-08-06
>
> We would like to thank the reviewers for their constructive feedback and for recognizing the strengths of our work. We appreciate the detailed insights and suggestions provided. Below, we address the key points raised in your review.
>
> > **W1: Lack of Clarification in Presentation**
>
> We acknowledge the need for better clarification and consistency in the presentation. We will fix these typos in the revision. We would like to clarify some of the points for your concern.
>
> > **W2: More Experiments Are Required**
>
> Thank you for your constructive advice. Following most of the existing works on VLMs, we report the same VLM benchmarks to examine the cross-modal ability for better comparison.
>
> We also understand and agree with your concern that CAL might affect performance on language-only tasks. To address your concern, **we experimentally verified that CAL did not interfere the performance on language-only tasks**. We further  conducted additional evaluations on language-only tasks including AGI-eval, BBH, and MMLU. These statistics show that CAL achieves comparable results, and even outperforms the baseline on BBH on LLaVA-15-13B.
>
> | Method         | AGI-eval | BBH    | MMLU  |
> | -------------- | :------: | :----: | :---: |
> | LLaVA-15-7B    | 38.14    | 40.78  | 50.56 |
> | LLaVA-15-7B + CAL | 37.65 | 40.33  | 50.63 |
> | LLaVA-15-13B   | 39.55    | 49.27  | 55.11 |
> | LLaVA-15-13B + CAL | 39.52 | 49.67 | 55.14 |
>
> As for hallucination issues, our method prioritizes visually correlated tokens to enhance cross-modal alignment. Nevertheless, since we maintain a minimal weight for all tokens, some contradictory tokens inevitably continue to impact the training process, which can potentially lead to hallucinations as baseline model does. Furthermore, it's important to recognize that LLMs inherently exhibit hallucination issues. Merely enhancing cross-modal alignment, although beneficial, is not a comprehensive solution. This challenge necessitates the adoption of additional techniques tailored to address this specific problem. Besides, CAL shows comparable performance on POPE with the baseline models, as presented in the Appendix in Table 7.
>
> > **W3: Lack of Originality**
>
> Thank you for your feedback. We would like to clarify the originality of both our motivation and methodology.
>
> For motivation, we are the first to study token discrepancy in the cross-modality alignment stage of existing VLMs, while [1] and [2] focus on language-only scenarios. We are glad to include them in the related works, but we would like to clarify the significant difference from these works. Our work and [2] are concurrent but use different techniques: [2] trains a token-level selective model using high-quality data, while we select tokens based on contrastive image inputs. Unlike [1] and [2], our motivation is to enhance cross-modal alignment by prioritizing visually correlated tokens rather than handling noise/hard tokens.
>
> For similarity with contrastive-decoding methods, we emphasize that contrastive learning is widely used but applied differently across scenarios. Our innovation lies in token-level training data selection by identifying visually correlated tokens in training labels using contrastive learning and utilizing logit differences for token loss reweighting.
>
> > **Q1: Trend in Figure 2a**
>
> Thank you for your insightful question. The trend shown in Figure 2a is a common phenomenon across various VLMs, including LLaVA-15-7B-PT, LLaVA-15-7B-SFT, and MGM-7B-SFT. The high cosine similarity in weight assignment distribution across these models on ShareGPT4V dataset, as shown below, supports this:
>
> | Model              | LLaVA15-7B-SFT | MGM-7B-SFT |
> | ------------------ | :------------: | :--------: |
> | LLaVA15-7B-PT      | 0.9470         | 0.9352     |
> | LLaVA15-7B-SFT     | -              | 0.9191     |
>
> > **Q2: Handling Split Semantic Words**
>
> Thank you for your question. We do not strictly distinguish or design special methods for subwords since CAL is orthogonal to word tokenization. Currently, there is no theoretical basis to decompose sub-word level interactions within a semantic word. We consider the logit change at the token level.
>
> > **Q3: Model's Ability to Predict Logit Changes**
>
> Thank you for your feedback. The current standard practice in training VLMs primarily aims to map image features to the text embedding space. Before this alignment stage, the CLIP ViT is pre-trained on a billion-level dataset, and the LLM is trained on a trillion-level dataset. These models are highly capable of encoding images and handling complex language tasks. In the first phase of VLM training, it is common practice to freeze the ViT and the LLM, and only train a projector to align their features. This step can be quite straightforward and quick. The figure in Section 2 comes from the pre-trained LLaVA-NeXT-13B, demonstrating that the ability to distinguish tokens emerges early in the alignment process.
>
>
> > **Q4: Training Cost and "Lightweight" Claim**
>
>
> For the concern of "lightweight". **Experimentally**, as discussed in the Appendix (Table 10), CAL adds only a 20% increase in training time, far from "doubling". **Theoretically**, the additional forward pass of CAL neither needs gradient back-propagation nor needs storing the intermediate activation values, which is both computation time friendly and memory efficient. Besides, the second forward pass does not need image tokens as the original forward process does, resulting in reduced sequence length and increased speed.
>
> ---
>
> We hope these clarifications and additional details adequately address your concerns. Thank you once again for your valuable feedback.

---

> > ### Comment · Reviewer_mF2S · 2024-08-12
> >
> > I appreciate the authors' response in addressing my questions and concerns. The explanation effectively clarifies the concern regarding training costs, particularly noting that the second forward pass does not require storing intermediate activations or performing backward propagation. However, the claim of a 20% increase in time is not scientifically grounded, as it can vary depending on the dataset used for training.
> >
> > My third question was not fully understood by the authors. The inquiry was not about the training recipe for MLLMs like LLaVA. In the early stages of training, one might assume the model lacks the ability to generate captions from images but the LLMs can perfectly generate text from a partial caption. The question is whether the gap in predicted logits with and without the image is accurate in the initial stage where the model lacks good captioning capacity. It might be more appropriate to apply this training during the instruction-tuning stage, where it is assumed the model can generate captions after pretraining. Another way to frame my question is: considering that language models typically use RoPE as a positional embedding, how does prepending image tokens affect logits given the altered positional embedding of text tokens in this scenario?
> >
> > Overall, while some concerns remain, particularly regarding the limited originality of the methodology in light of prior works, I find the work interesting and believe it may inspire future research. As such, I have raised my score to 5.

---

> > > ### Author Response · Authors · 2024-08-14
> > >
> > > Thank you for your detailed feedback and for raising your score. We appreciate your recognition of our work's potential and your valuable suggestions.
> > >
> > > Regarding the computational cost, we can preprocess and store the CAL weights using the trained model offline, which will not increase the cost in subsequent training. This is one of our future improvement directions.
> > >
> > > In the pretraining stage, since the number of trainable parameters (only projector) is limited and there is a minimum weight for all text tokens, inaccuracies in predicting logits in the early stages do not significantly impact the overall performance. As shown in Table 3, applying CAL in both pretrainig and finetuning stages yields the best performance.
> > >
> > > Regarding the issue of positional embeddings, in the current version, we ensure that the positional embeddings for text tokens are consistent in both two scenarios.
> > >
> > > Once again, thank you for your valuable insights and support. We are committed to addressing these issues in our future work.

---

### Official Review · Reviewer_ewhK · 2024-07-09

**Soundness:** 2
**Presentation:** 3
**Contribution:** 2
**Rating:** 6
**Confidence:** 4

**Summary:**

This paper introduces Contrastive Alignment (CAL), a straightforward yet effective re-weighting strategy designed to improve multimodal alignment. Specifically, the authors propose contrasting image inputs and calculating the differences in prediction logits for each text token to determine its training weights. Extensive experiments are conducted to demonstrate the effectiveness of CAL across various tasks, providing a robust validation of the proposed approach.

**Strengths:**

1. The Contrastive Alignment (CAL) proposed in this paper is an efficient method for multimodal alignment, particularly beneficial when training data is heavily contaminated with noise. The dynamic adjustment of training weights represents a lightweight yet innovative solution to this problem.
2. The experimental section of this paper is thorough. It includes extensive quantitative analysis and demonstrates the performance of CAL across a broad range of scenarios. These results not only allow for a deep understanding of the CAL method but also provide valuable insights into its practical applications.

**Weaknesses:**

1. While CAL is innovative for aligning multimodal data, its robustness and comprehensiveness in complex task settings, such as multimodal question answering (VQA), appear limited. The method ensures correct alignment but may inadvertently impair other model capabilities, such as reasoning ability or retention of original world knowledge. This issue is critical in VQA scenarios where data often include elements not directly related to image content, potentially conflicting with it and incorporating extensive external knowledge. The application of CAL in such contexts might suppress training on these critical aspects, introducing uncontrollable biases and worsening model performance. Although CAL shows promise in simpler tasks like captioning and grounding, its performance in VQA tasks is inconsistent, and its effectiveness in addressing hallucination issues, such as in the POPE dataset, is negligible.
2. CAL’s effectiveness appears contingent upon the pre-existing performance of the underlying multimodal model. Since it calculates visually correlated weights based on the model itself, any hallucinations or errors present in the base model are not only perpetuated but potentially exacerbated by the CAL method. This limitation restricts CAL’s applicability and caps its performance, as there is no mechanism within CAL to correct these amplified errors.

**Questions:**

1. Why MGM-HD-13B perform worse that -7B model in caption tasks as shown in Tab-2 in the paper ?
2. The visually correlation weights in CAL are smoothed using average pooling with a window size of W. How is W determined? Is this parameter set consistently across all datasets, or does it require adjustment for each training run to optimize performance?
3. The paper mentions that CAL adds minimal computational overhead compared to other data scaling strategies. However, could the authors provide more quantitative examples or an intuitive comparison? It seems that processing very long texts could significantly increase the overhead.

---

> ### Author Rebuttal · Authors · 2024-08-06
>
> We would like to thank the reviewers for their constructive feedback and for recognizing the strengths of our paper. We appreciate the detailed insights and suggestions provided. Below, we address the key points raised in your review.
>
> > **W1: Robustness in Complex Task Settings Like VQA**
>
> Actually, aligning with previous works, we have conducted extensive validation on various VQA tasks, as shown in Table 1(Doc, Chart, Text VQA, mmstar, mmtbench and etc). And our methods surpass the baseline in most cases.
>
> As for the retention of original world knowledge, we present the performance on OKVQA[1], which is used to test the model's ability to answer questions requiring external knowledge beyond what is explicitly shown in the image. The CAL models exceed baseline models in most cases.
>
> | Model  | Baseline | +CAL |
> |-------|-------------|-----------------|
> | LLaVA-15 7B | 56.46       | 58.77           |
> | LLaVA-15 13B| 53.09       | 56.20           |
> | LLaVA-NEXT 7B | 56.08       | 55.26         |
> | LLaVA-NEXT 13B| 55.09       | 55.69         |
>
> We also evaluate the pure text capability of the baseline and CAL models. Among the tests used, BBH is specifically employed to assess reasoning ability. There are no significant differences compared to the baseline.
>
> | Method         | AGI-eval | BBH(For reasoning)| MMLU |
> | -------------- | :------: | :----: | :---: |
> | LLaVA-15-7B    | 38.14    | 40.78  | 50.56 |
> | LLaVA-15-7B + CAL | 37.65 | 40.33  | 50.63 |
> | LLaVA-15-13B   | 39.55    | 49.27  | 55.11 |
> | LLaVA-15-13B + CAL | 39.52 | 49.67 | 55.14 |
>
>
> As for hallucination issues, our method prioritizes visually correlated tokens to enhance cross-modal alignment. Nevertheless, since we maintain a minimal weight for all tokens, some contradictory tokens inevitably continue to impact the training process, which can potentially lead to hallucinations as baseline model does. Furthermore, it's important to recognize that LLMs inherently exhibit hallucination issues. Merely enhancing cross-modal alignment, although beneficial, is not a comprehensive solution. This challenge necessitates the adoption of additional techniques tailored to address this specific problem.
>
> > **W2: Dependency on Base Model Performance and Amplification of Existing Errors**
>
> Thank you for your observation regarding CAL's dependence on the base model's performance.
>
> While it is true that CAL could potentially amplify errors, we would like to clarify that **this scenario is rare**. The amplification of errors only occur when there are both **hallucinations in the training dataset** and **such hallucinations are also captured by CLIP**, which is rare in practice. Moreover, as demonstrated in our **experiments on the POPE** (for hallucination evaluation) dataset, our method achieves comparable performance with the baseline.
>
> | POPE Performance     | Baseline | +CAL  |               | Baseline | +CAL  |
> |----------------------|----------|-------|---------------|----------|-------|
> | MGM 7B               | 85.7     | 87.5  | LLaVA-15 7B   | 86.2     | 85.5  |
> | MGM 13B              | 86.2     | 86.4  | LLaVA-15 13B  | 85.7     | 85.8  |
> | MGM-HD 7B            | 85.7     | 87.0  | LLaVA-NEXT 7B | 86.8     | 86.7  |
> | MGM-HD 13B           | 86.3     | 86.4  | LLaVA-NEXT 13B| 87.2     | 87.2  |
>
>
> > **Q1: Performance Discrepancy Between MGM-HD-13B and MGM-HD-7B**
>
> The performance discrepancy arose from the initial zero-shot evaluation approach in MGM, where results were largely influenced by output style. We retrained the MGM-HD model using the LLAVA-15 dataset, which includes training data from both COCO and TextCaps. The retrained results showed that the performance trend in MGM aligns with the LLAVA models, resolving the discrepancy.
>
> | Model          | COCOCaps | TextCaps | Model          | COCOCaps | TextCaps |
> | -------------- | -------- | -------- | -------------- | -------- | -------- |
> | MGM 7B         | 107.54   | 108.17   | MGM 7B + CAL   | 110.33   | 113.33   |
> | MGM 13B        | 113.00   | 112.47   | MGM 13B + CAL  | 113.91   | 118.10   |
>
> > **Q2: Smoothing Window Size (W) Determination**
>
> We fixed the window size (W) to 3 in all our experiments to smooth the weight distribution. An additional experiment where we removed the average pooling step showed slightly inferior performance, supporting our chosen.
>
> We will include these findings and a discussion on the fixed window size in the revised paper.
>
> | Benchmark      | ChartQA | Docvqa|  SQA-I | COCO Caption| TextCaps| OCRBench | Refcocog val|
> | -------------- | --- | --- | --- | --- | --- | --- | --- |
> | w/ AvgPool     | 67.2 | 80.1 |  71.5 | 120.6 | 124.4 | 574 | 80.4 |
> | w/o AvgPool    | 66.3 | 79.5 |  72.4 | 116.7 | 123.8 | 581 | 79.5|
>
> > **Q3: Computational Overhead of CAL**
>
> As detailed in Table 10, CAL generally introduces an additional computational cost of approximately 20%. This overhead is primarily due to **an extra forward pass without gradient computation**.
>
> In exceptionally long texts scenarios, the original forward and backward process also need to deal with long sequences. Therefore, although the absolute overhead increases in such scenarios, the relative increase compared to shorter samples remains consistent.
>
> ---
>
> [1] OK-VQA: A Visual Question Answering Benchmark Requiring External Knowledge

---

> > ### Comment · Reviewer_ewhK · 2024-08-10
> >
> > Thanks for the response, most of my concerns have been addressed. However, for an algorithm that introduce additional computational costs, mere performance enhancements in "most" scenarios are insufficient, particularly as we observe that the CAL method results in performance degradation in the LLaVA-Next model. This phenomenon could be anticipated given the lack of interpretability of CAL in complex settings, or as discussed in W2, the introduction of uncontrollable biases stemming from inherent limitations of the base model. Despite its flaws, this work challenges the dominant paradigm of pure supervised learning in multimodal training, making it a valuable contribution to the field. In recognition of its potential, I will raise my score.  I urge the authors to provide more quantitative analyses in future version of the paper, such as exploring the weight distribution highlighted by Reviewer DwVB, or presenting scenarios where CAL fails to enhance performance, to deepen our understanding of these issues.

---

> > > ### Author Response · Authors · 2024-08-14
> > >
> > > Thank you for your detailed feedback and for raising your score. We appreciate your recognition of our work's potential and your valuable suggestions.
> > >
> > > Regarding the computational cost, we can preprocess and store the CAL weights using the trained model offline, which will not increase the cost in subsequent training. This is one of our future improvement directions.
> > >
> > > In future versions of our paper, we will add more analyses, including exploring weight distribution and scenarios where CAL may not enhance performance. In our subsequent work, we will strive to address the uncertainties introduced by the base model.

---

### Official Review · Reviewer_tsLt · 2024-07-11

**Soundness:** 4
**Presentation:** 3
**Contribution:** 4
**Rating:** 7
**Confidence:** 4

**Summary:**

This paper points out that parts of samples in the broadly used datasets contain visually contradictory text tokens. To mitigate the sub-optimal cross-modal alignment in VLMs, this proposed method is to assign distinct contributions for each text token based on its visual correlation.

**Strengths:**

1. The proposed method is easy to implement and effective. The idea of assigning distinct weights for each token is intuitive.
2. This paper is well organized and provides clear preliminaries, helping readers understand the method.
3. CAL achieves consistent and solid performance across different benchmarks.

**Weaknesses:**

Please see the Questions

**Questions:**

1. It’s unclear how to calculate the prediction logit distribution o without input I for the VLM like CLIP. Is this method not applicable to the pure image-text matching VLM like CLIP?
2. There’s no l in Equation 4. Does it mean W in Sec. 3.1 implementation details “we set l in Equation 4 to 3 for all experiments”?
3. For Figure 3, it is better to describe what the dashed line represents as well.

---

> ### Author Rebuttal · Authors · 2024-08-06
>
> We would like to thank the reviewers for their positive feedback and for highlighting the strengths of our work. We appreciate the detailed insights and suggestions provided. Below, we address the key points raised in your review.
>
> > **Q1: How to Use on CLIP**
>
> Thank you for your question regarding the calculation of the prediction logit distribution (o) without input (I) for Vision-Language Models (VLMs) like CLIP.
>
> For models like CLIP, we can assess the image-text relation by evaluating the change in CLIP score between the original and the noise-added images. This approach aligns with the essence of our method, which is focused on contrastive learning.
>
> > **Q2: There’s No 'l' in Equation 4. Does It Mean 'W' in Sec. 3.1?**
>
> Thank you for pointing out this issue. We acknowledge the typo in our manuscript and will correct it in the revised version of the paper.
>
> > **Q3: Dashed Line in Figure 3**
>
> The dashed line represents the asymptote. We will update the figure description in the revised version of the paper to clearly indicate this and prevent any further confusion.
>
> ---
>
> We hope that these clarifications and additional details adequately address your concerns. Thank you once again for your valuable feedback.

---

> > ### Comment · Reviewer_tsLt · 2024-08-13
> > **Response to rebuttal by authors**
> >
> > Thanks for the response. The Answer for Q1 is still a vague idea in its infancy. In future work, the authors may implement the CAL on CLIP by using "noise-added images" to decide the token scores and see if it works. I choose to keep my original positive rating.

---

> > > ### Author Response · Authors · 2024-08-14
> > >
> > > Thank you for your constructive feedback and for maintaining your positive rating. We appreciate your suggestion and will explore implementing CAL on CLIP with noise-added images in future work.

---

> ### Comment · Area_Chair_om21 · 2024-08-13
>
> Hi Reviewer  tsLt,
>
> Could you take a look at the authors' rebuttal and finalize your rating?
>
> Thanks,
> AC

---

### Official Review · Reviewer_DwVB · 2024-07-13

**Soundness:** 2
**Presentation:** 3
**Contribution:** 2
**Rating:** 6
**Confidence:** 3

**Summary:**

This study proposes a reweighting strategy, namely contrastive alignment, to enhance model learning on visually correlated tokens. Specifically, the authors divided the tokens into three sub-groups, virtually related, non-related, and contradictory, and assigned different weights to each group. The weight is calculated based on the predicted contrastive logit (w/o visuals) and further post-processed by clamping and average pooling. This study is evaluated on vision question answering, image caption, and grounding tasks using two foundation model structures, LLaVA and Mini-Gemini.

**Strengths:**

-- The paper is well-written.

-- Deploying a reweighting strategy on different types of tokens seems feasible for performance improvement.

-- The division of token groups is straightforward yet reasonable.

**Weaknesses:**

-- This study does not compare with any baseline models within the topic of resisting noisy tokens, such as [*] and [**].

-- In [*], a study with similar aims was presented. In particular, they also studied the image-token consistency. Thus, detailed baseline comparisons, including overall performance and token consistency evaluation/comparison, are expected.

-- The authors claim that the proposed strategy is simple yet efficient. However, it may rely upon several strong assumptions, such as high capability requirements of the pre-trained model. These assumptions are not presented, discussed, and validated in the paper.

-- Weight distribution on different token sub-groups is not presented. This would contribute to the validation of weighting.

[*] Gou, Yunhao, et al. "Leveraging per image-token consistency for vision-language pre-training." Proceedings of the IEEE/CVF Conference on Computer Vision and Pattern Recognition. 2023.

[**] Wu, Cheng-En, et al. "Why Is Prompt Tuning for Vision-Language Models Robust to Noisy Labels?." Proceedings of the IEEE/CVF International Conference on Computer Vision. 2023.

**Questions:**

See above weakness.

**Limitations:**

As I can see, this paper has no potential negative societal impact.

---

> ### Author Rebuttal · Authors · 2024-08-06
>
> We would like to thank you for your constructive feedback and highlighting the strengths of our work. We appreciate the detailed insights and suggestions provided. Below, we address the key points raised in your review.
>
> > **W1: Lack of Baseline Comparisons**
>
> We are happy to include these papers in the related work, however, we would like to clarify that these works might not be suitable for comparison as baseline methods in this paper. First of all, the two methodologies mentioned are primarily **designed for visual-language representation tasks, focusing on classification and retrival tasks**. However, CAL is designed for generation task, so there are no common benchmarks for comparision. Secifically, Gou et al. (2023) use a method involving masked images and a **BERT-like** approach to identify salient tokens, which is not applicable to the **causal decoder-only large language models** used in our method. Wu et al. (2023) focus on enhancing the CLIP pretraining scheme, which is different from the multimodal generation tasks we address. Secondly, our method focuses on ensuring better alignment between visual and textual feature spaces by prioritizing visually correlated tokens. While our method does reduce the influence of noisy tokens, its primary aim is to diminish the impact of irrelevant tokens, which constitute a much larger proportion. Therefore, addressing noise tokens is not its central goal.
> > **W2: Strong Assumptions**
>
> Thank you for pointing out this concern regarding the assumptions in our proposed strategy.
>
> 1. The basic assumption of CAL is the **high image-text matching capability of Vision Transformer** and a **well-trained Large Language Model**, which are already facts in current structures of Generation VLMs (like LLaVA). The current standard practice in training VLMs primarily aims to map image features to the text embedding space. Before this alignment stage, the CLIP ViT is pre-trained on a billion-level dataset, and the LLM is trained on a trillion-level dataset. They are of great capability to encode images and deal with complex language tasks. In comparison, the data (about 1M samples in LLaVA) used during the VLM training stage mainly serves to align the feature spaces of the two models and to enable the model to answer image-related questions.
>
> 2. Another assumption of CAL is that **after training a simple projector only, the VLM is capable of distinguishing visually correlated tokens**. In the first phase of VLM training, it is common practice to freeze the ViT and the LLM, and only train a projector to align their features. In the second phase, we finetune the model using high-quality data to enable it to answer image-related questions. To validate this assumption, **we finetune CAL models using two types of pre-trained versions: one with the original pre-trained model (only train the projector) and one with the fully trained baseline model (after the finetuning phase)**. We then compared the performance of these two versions and found no significant differences in performance. We will include a more detailed discussion of these assumptions and their validation in the revised version of the paper to address this concern explicitly.
>
> | Model | Pretrain | ChartQA | Docvqa | SQA-I | COCO Caption | TextCaps | OCRBench | Refcocog val |
> | --- | --- | --- | --- | --- | --- | --- | --- | --- |
> | Baseline(LLaVA-Next-13B) | Original Pretrain | 63.8 | 78.4 | 71.8 | 118.5 | 118.2 | 553 | 79.8 |
> | CAL | Original Pretrain | 67.2 | 80.1 | 71.5 | 120.6 | 124.4 | 574 | 80.4 |
> | CAL | Baseline Pretrain | 66.3 | 79.5 | 72.4 | 116.7 | 123.8 | 581 | 80.2 |
>
>
> > **W3: Weight Distribution Not Presented**
>
> Thank you for your insightful comment on the weight distribution across different token sub-groups.
>
> We appreciate your recognition of our token classification approach. However, it is important to note that there is no existing labeled dataset that directly identifies these three types of tokens. Given the impracticality of manually annotating a large volume of data at the token level within a short time frame, we utilized the RLHF-V[1] dataset, **which includes annotations for incorrect words in captions**.
>
> Using this dataset, we conduct a statistical analysis and find that **the average logits difference (diff) for erroneous tokens is significantly lower than that for correct tokens**. This finding supports our hypothesis about the effectiveness of our weighting mechanism. We will include these details and present the weight distribution analysis in the revised version of the paper to provide a more comprehensive validation of our weighting approach. It is worthy to mention that since this dataset is specially collected, the number of false tokens is much larger than other tokens. (LLaVA-15)
>
> | Token Type | Token Num | avg_diff_value (7B model) | avg_diff_value (13B model) |
> | --- | --- | --- | --- |
> | False token | 344197 | 0.163 | 0.277 |
> | Correct token | 76783 | 2.271 | 2.10 |
>
> ---
>
> We hope that these clarifications and additional details adequately address your concerns. Thank you once again for your valuable feedback.
>
> [1] Rlhf-v: Towards trustworthy mllms via behavior alignment from fine-grained correctional human feedback

---

> > ### Comment · Reviewer_DwVB · 2024-08-13
> >
> > The authors partially addressed my concerns. So, I increased my rate.

---

> > > ### Author Response · Authors · 2024-08-14
> > >
> > > Thank you for your feedback and for raising your score. We appreciate your recognition of our work's potential and your valuable suggestions.

---

> ### Comment · Area_Chair_om21 · 2024-08-13
>
> Hi Reviewer  DwVB,
>
> Could you take a look at the authors' rebuttal and finalize your rating?
>
> Thanks,
> AC

---

### Decision · Program_Chairs · 2024-09-25

**Decision:**

Accept (poster)

**Comment:**

This paper introduces an approach for multimodal alignment through contrastive alignment by assigning different weights for different tokens. All reviewers agree that the paper is easy to follow and well-written, and the idea is interesting. The authors' rebuttal addressed misunderstandings in terms of experiments and all reviewers recommends acceptance. The AC thus recommends acceptance and encourages the authors to revise the paper accordingly.